# Simulation of Electrical and Thermal Properties of Granite under the Application of Electrical Pulses Using Equivalent Circuit Models

**DOI:** 10.3390/ma15031039

**Published:** 2022-01-28

**Authors:** Kyosuke Fukushima, Mahmudul Kabir, Kensuke Kanda, Naoko Obara, Mayuko Fukuyama, Akira Otsuki

**Affiliations:** 1Graduate School of Engineering Science, Department of Mathematical Science and Electrical-Electronic-Computer Engineering, Tegata Campus, Akita University, 1-1 Tegata Gakuen Machi, Akita 010-8502, Japan; m8020422@s.akita-u.ac.jp (K.F.); m8021405@s.akita-u.ac.jp (K.K.); obara@gipc.akita-u.ac.jp (N.O.); 2Graduate School of Engineering Science, Cooperative Major in Life Cycle Design Engineering, Tegata Campus, Akita University, 1-1 Tegata Gakuen Machi, Akita 010-8502, Japan; mayuko@gipc.akita-u.ac.jp; 3Ecole Nationale Supérieure de Géologie, GeoRessources, UMR 7359 CNRS, University of Lorraine, 2 Rue du Doyen, Marcel Roubault, BP 10162, 54505 Vandoeuvre-lès-Nancy, France; akira.otsuki@univ-lorraine.fr; 4Waste Science & Technology, Luleå University of Technology, SE 971 87 Luleå, Sweden

**Keywords:** electric field, temperature distribution, conductivity, dielectric constant, liberation, hard-rock, mineral distribution

## Abstract

Since energy efficiency in comminution of ores is as small as 1% using a mechanical crushing process, it is highly demanded to improve its efficiency. Using electrical impulses to selectively liberate valuable minerals from ores can be a solution of this problem. In this work, we developed a simulation method using equivalent circuits of granite to better understand the crushing process with high-voltage (HV) electrical pulses. From our simulation works, we calculated the electric field distributions in granite when an electrical pulse was applied. We also calculated other associated electrical phenomena such as produced heat and temperature changes from the simulation results. A decrease in the electric field was observed in the plagioclase with high electrical conductivity and void space. This suggests that the void volume in each mineral is important in calculating the electrical properties. Our equivalent circuit models considering both the electrical conductivity and dielectric constant of a granite can more accurately represent the electrical properties of granite under HV electric pulse application. These results will help us better understand the liberation of minerals from granite by electric pulse application.

## 1. Introduction

Minerals are familiar to us in our daily lives and are used in various fields (e.g., ceramics, pharmaceuticals, cosmetics) [1]. In order to extract minerals from rocks, a combination of mechanical comminution methods of crushing and grinding (e.g., ball mill) based on compression, impact and shearing mechanisms are generally used [2]. The energy efficiency of these methods is as low as 1%. Especially for hard rocks, such as granite, the mechanical grinding is inefficient due to tool wear and the increase in cost [3]. In addition, the energy used for comminution is very large, accounting for 2–4% of the world energy consumption [4,5]. Therefore, a new method for size reduction and selectively liberating minerals from rocks using high-voltage electric pulses has been developed [2]. Electric pulse liberation of minerals has been reported as a solution for low energy efficiency in comminution [6,7,8].

There are two types of electrical disintegration methods, and they are (1) Electro Hydraulic Disintegration (EHD) and (2) Electrical Disintegration (ED). Figure 1 shows a schematic of the EHD and ED methods. For (1) EHD, an upper/high voltage (HV) electrode and a ground electrode are placed above the object in a liquid and a shock wave is generated for the selective liberation of minerals (Figure 1a). (2) The ED method is also applied in a liquid but the upper/HV electrode is placed directly on or very close to the object to be disintegrated (Figure 1b). The ground electrode is placed below the object and when the electric pulse is applied, a large electric current is generated and flows through the object and thus leads to its disintegration. Water, due to its low electrical conductivity, is usually used as a liquid. For the EHD method, the voltage is applied only to the rock because the breakdown voltage of the rock is lower than the electric field strength of the water when the rise time of the electric pulse is less than 500 ns [9]. In the EHD method, lightning impulses are applied to two electrodes in close proximity, and the compressive force generated by the shock wave is used for crushing. In the ED method, the upper/HV electrode is placed close to the sample, and an instantaneous large current is applied to the inside of the sample to crush it [9]. The ED method is used in this study because of its potential for selective grinding and its ability to weaken the rock and reduce the energy consumption for grinding hard rocks [10,11].

The ED method has been applied to the separation of precious minerals such as diamonds and emeralds [11]. In addition, the effects of electric pulse application on rocks have been studied based either on the electric conductivity or dielectric constant of rocks experimentally or simulation [11,12,13]. Walsh et al. (2020) performed simulation of hard rocks under HV pulse application considering the microstructures (i.e., distribution of minerals) of the rocks and they described about the change in minerals of the rocks under HV application. Li et al. (2018) showed how the *I*-*V* properties of granite changed under HV applications with different types of electrode spacing. However, the effects of electric pulses on rocks are still largely empirical, and our work is the very first study that considered both conductivity and dielectric constant as far as the authors are aware of.

The purpose of this study is to obtain better knowledge of granite comminution and selective liberation under high voltage pulses. Granite was selected as an example of hard rock. For this purpose, we developed an equivalent circuit model of granite based on both conductivity and dielectric constant, considering the minerals and voids in the granite, and applied a high electric field to the granite model to simulate electrical characteristics under HV applications. Specifically, our equivalent circuit model was created using a C# program, and the electric field distribution in granite was calculated using the circuit simulation software LTspice. Since rocks consist mostly of dielectric materials, our simulation works can be significantly helpful to understand the behavior of rocks (i.e., granite) under HV electric pulse, as the circuit models were proposed considering both electric and dielectric properties of granite.

## 2. Simulation Methods

### 2.1. Simulation Model

The simulation was performed for a well-known hard rock, i.e., granite composed of quartz, plagioclase, k-feldspar, and biotite [12]. In this study, a granite specimen with a cube of 1 mm on each side was considered, assuming that the entire rock is divided into an arbitrary number of smaller cubes to create an equivalent circuit model. In general, the dimension of a rock (i.e., granite in this work) is larger than that of our simulation works. We assumed the granite as a small cube of 1-mm size on each side in order to avoid the complexity of the work. As the electrical properties such as conductivity and relative permittivity do not vary with their sizes, it is quite reasonable to consider a smaller size for simulation works. By comparing larger and smaller sizes of the rocks, the results will show similar trends and this point was reconfirmed from our simulation results, similar to what was reported in the literature [12]. To obtain accurate results, we assumed 1000 parts of smaller cubes which make the cube of 1-mm edge length of granite, as shown in Figure 2a. The electrical properties of minerals can be represented by a parallel circuit of resistors and capacitors (Figure 2b). The resistance of minerals in the equivalent circuit model is calculated from the following equation:(1)$$R=\frac{1}{\sigma }\frac{L}{S}$$
where *R* is the resistance, σ is the electrical conductivity, *L* and *S* represent the length of the segmented mineral (0.1 mm) and the surface area of the segmented mineral (0.01 mm^2^), respectively. Capacitance is expressed by the following equation:(2)$$C=\varepsilon \frac{S}{L}$$
here, *C* is capacitance and *ε* is dielectric constant. The equivalent circuit was created by assigning the minerals contained in the granite to the divided cubes, calculating the resistance and capacitor values for each mineral, and constructing the circuit. As an example, Figure 2 shows the equivalent circuit model for a 10 × 10 × 10 partitioned cube. The simulation work was performed for a needle-flat electrode as shown in Figure 2a. From the knowledge of electric circuits, it can be understood that the electric field strength is mainly applied in the vertical direction. The electric field strength in the horizontal direction was neglected because the values were less than 3% compared to the ones in vertical direction. Thus, in this paper, we have shown the electric field strengths in the vertical direction only. The potential difference for each circuit element was calculated by dividing the values by the distance (i.e., 0.1 mm). Then, the electric field was calculated for each smaller cube and the data were used to draw the 3D mapping of the electric field distribution with MATLAB (R2016a, The MathWorks, Inc., Natick, MA, USA). Other simulation results were also used to draw 3D images by MATLAB.

### 2.2. Composition of Granite

The mineral composition of a typical granite is referred to the information available in [9], as shown in Table 1. The distribution of each mineral in the granite was determined by a random placement program based on the compositional ratio using the programming software C#.

### 2.3. Electrical and Thermal Properties of Minerals

The current ***J*** per unit area flowing through a rock is expressed by Ohm’s law:(3)$$J=\sigma E+\frac{\partial D}{\partial t}=\sigma E+\varepsilon \frac{\partial E}{\partial t}$$
where ***E*** is the electric field, and *ε* is the dielectric constant [14]. From Ohm’s law, the current distribution in a rock is affected by the electrical conductivity and dielectric constant. The relationship between temperature and electrical conductivity of minerals ($\sigma_{s}$) is empirically expressed by the following equation [12]:(4)$$\sigma_{s}=Aexp\left(\frac{-B}{k_{b}T}\right)$$
where *A* and *B* are constants, *T* is the absolute temperature, and $k_{b}$ is Boltzmann’s constant (8.618 × 10^−5^ eV/K). The constants *A* and *B* of the granite rocks are shown in Table 2 [12], and the electrical conductivity of the minerals as a function of temperature is shown in Figure 3. The values of conductivity in Figure 3 were calculated for each single mineral using Equation (4). It is worth mentioning that the conductivity values of minerals (i.e., Figure 3) are surely different from the conductivity values of a mix solution (i.e., for NaCl solution of Equation (7)). At low temperature, the electrical conductivity of minerals is mainly affected by the water content in the fractures and pores. Sinmyo and Keppler reported the following empirical equation for NaCl solutions [15]:(5)$$\log_{10}\sigma_{Brine}=-1.7060\;-\frac{93.78}{T}+0{.8075\log}_{10}c+3{.0781\log}_{10}{\rho +\log}_{10}\left(\lambda_{0}\left(\rho ,T\right)\right)$$
where *c* is the mass percent concentration of NaCl, *ρ* is the mineral density, and $\lambda_{0}$ is the molar conductivity of NaCl in water at unlimited dilution. Here, $\lambda_{0}$ is approximated by the following equation:(6)$$\lambda_{0}=1573\;-1212\rho +\frac{537,062}{T}-\frac{208,122,721}{T^{2}}$$

Equation (5) can only be applied to temperatures above 100 °C, but the following equation can be used to approximate the electrical conductivity below 150 °C [16]:(7)$${\sigma \;}_{f}{=\sigma }_{ref}\frac{T-251.65}{T_{ref}-\;251.65}$$
where $\sigma_{ref}$ is the electrical conductivity at the reference temperature, and the unit of $\sigma_{ref}$ is K. In this simulation, the $\sigma_{ref}$, T and *c* was set to 150 °C (423.15 K), 20 °C (293.15 K), and 3 wt%, respectively [12,15]. Note that the temperature variation occurs instantaneously and we used Equation (17) to calculate the temperature of the granite under HV impulse application.

The electrical conductivity for a composite mixture of solid and fluid can be expressed using a modified Archie’s law proposed by Glover [17]:(8)$$\sigma_{mix}{=\sigma }_{s}{\left(1\;-\emptyset_{f}\right)}^{p}{+\sigma }_{f}\emptyset_{f}{}^{m}$$
where $\sigma_{s}$ is the electrical conductivity of a solid, $\sigma_{f}$ is the electrical conductivity of a fluid, $\emptyset_{f}$ is the volume fraction of a fluid, *m* is Archie’s exponent, and *p* is determined by Equation (9).
(9)$$p=\frac{\log\left(1-\emptyset_{f}{}^{m}\right)}{\log\left(1-\emptyset_{f}\right)}$$

The value of Archie’s index (*m* = 1.5) was used in this calculation and obtained from [12].

As shown in Equation (1), the value of dielectric constant is important in considering the current distribution within the minerals composing a rock. The volume of the fluid (considered equivalent to the void volume) and the dielectric constant of each mineral are shown in Table 3 [12,13].

### 2.4. Applied Voltage

In the case of ED crushing, the electrodes are placed directly on the rock and the voltage is applied, and thus, the mineral crushing is theoretically possible even without using lightning impulses. However, in practice, the application of high voltage for a long period needs a large amount of electricity and lowers the energy efficiency. Due to this point, it is reasonable to use lightning impulses for mineral crushing. A lightning impulse (0.3 μs/50 μs) was used as the input voltage applied to the granite in this work. The peak value of impulse voltage was at the front time (i.e., 0.3 μs = 300 ns), and 50% of the peak voltage was at the tail time (i.e., 50 μs). In this work, the lightning impulse was simulated and applied to the granite by the circuit simulator LTspice (Figure 4). Fujita et al. (1999) explained that the voltage was applied only to the rock when the rise time of the applied voltage was less than 500 ns in the ED method [9]. Therefore, in this study, the applied voltages were: (a) 100% rise, the maximum voltage 20 kV at a rise time of 300 ns, and (b) 50% rise, 50% of the maximum voltage 10 kV at 38 ns. The electric current *I* was also calculated and plotted within the same graph of applied voltage (i.e., Figure 4). The maximum current (i.e., 2.55 mA) was observed at 2.34 ps. At the maximum value of the lightning impulse voltage (i.e., 20 kV), the value of the current was 1.53 mA at 0.3 μs. After 0.3 μs, the value of the current gradually decreased to 0.75 mA at 50 μs.

### 2.5. Temperature in the Rock

Assuming a rock is regarded as a dielectric material, we will discuss the electric phenomena when an AC input voltage *v* with angular frequency *ω* is applied on the rock. Figure 5 shows the equivalent circuit for a typical dielectric material with AC input voltage. The AC analysis for this circuit can be explained with the use of vector analysis. The input voltage was chosen as the reference vector. The effective value of voltage *v* is *V* and the vector expression we used is the symbol of $\dot{V}$. For the circuit shown in Figure 5a, the total electric current $\dot{I_{p}}$ can be expressed by a vector sum of $\dot{I_{C}}$ and $\dot{I_{R}}$. Here, $\dot{I_{C}}$ represents the displacement current and $\dot{I_{R}}$ represents the loss current of the granite. $\dot{I_{C}}$ and $\dot{I_{R}}$ can be explained by Equations (10) and (11), respectively.
(10)$$\dot{I_{C}}=j\omega C\dot{V}$$
(11)$$\dot{I_{R}}=\frac{1}{R}\dot{V}$$

Here, *ω* is angular frequency of the AC voltage *v*. The sum of the electric current, i.e., $\dot{I_{P}}$ can be found by adding Equations (10) and (11).
(12)$$\dot{I_{P}}=\dot{I_{R}}+\dot{I_{C}}=(\frac{1}{R}+j\omega C)\dot{V}$$

The amount of heat per unit time generated by the applied voltage and current flowing through the granite (i.e., electric power) can be calculated by the following equation:(13)$$\frac{\partial q}{\partial t}{=VI}_{P}{\cos\theta =VI}_{R}=\frac{1}{R}V^{2}$$
where *q* is the heat generated, *V* is the potential difference and *θ* is the phase angle between the voltage and electric current $\dot{I_{P}}$. Assuming the phase angle between $\dot{I_{C}}$ and $\dot{I_{P}}$ is *δ*, the relationship in between *θ* and *δ* can be expressed by the following equation,
(14)$$\theta +\delta =\frac{\pi }{2}$$

From Figure 5b, the dielectric loss tan*δ* can be expressed by the following equation.
(15)$$\tan\delta =\frac{\dot{I_{R}}}{\dot{I_{C}}}=\frac{1}{R\omega C}$$

The values of resistance *R* and capacitance *C* were calculated for each mineral as shown in Table 4. The tan*δ* was also calculated at 20 kHz as shown in Table 4. From Equation (15), it is understood that the dielectric loss (i.e., tan*δ*) changes with frequency. The value of tan*δ* was 175 for quartz (see Table 4), but it can be changed to 1.17 at the frequency of 3 MHz. The temperature change caused by this heat can be expressed using a thermal resistance defined by the following equation:(16)$$R_{j}=\frac{1}{k}\frac{L}{S}$$
where $R_{j}$ is the thermal resistance and *k* is the thermal conductivity. The relationship between temperature change and heat is expressed by the following equation:(17)$$\frac{\partial q}{\partial t}=\frac{T_{1}{-T}_{2}}{R_{j}}$$
where *T*_1_ and *T*_2_ are the temperatures before and after the application of high voltage (i.e., heat generation) to the granite, respectively. The thermal conductivity of each mineral is summarized in Table 5 [18]. Equation (17) was used to calculate the temperature of granite under the HV application to be discussed later.

## 3. Results and Discussion

### 3.1. Distribution of Minerals in Granite

Using a program developed in C#, the distribution of minerals was determined, based on the mineral composition of the granite shown in Table 1. The layout information of the granite rock is shown in Figure 6a,b. Since the placement information is three-dimensional, we displayed in a cross-sectional view where the electrode is assumed to be placed, as shown in Figure 6b that is also a reference to Figure 6c–h in terms of mineral and electrode placement. The mineral distribution was chosen randomly meeting the conditions of Table 1 and Table 3. Some works [19,20] reported mineral distributions arranged with Voronoi networks in order to describe any random distribution of particles, minerals, etc. Though our simulation works used a 5 times larger void, comparing to their works, the proportion of void and minerals was maintained as described in Table 3.

### 3.2. Simulation Results

The rock was divided into 1000 small blocks (10 × 10 × 10), and minerals were placed in the blocks based on the mineral composition of the granite rock (Table 1). An equivalent circuit model was created based on the mineral distribution information (Figure 6b), and was used to investigate the electric field distribution in the granite when an electric pulse was applied to the granite. The applied voltage was set to reach the maximum voltage 20 kV at around 300 ns, as shown in Figure 4. The heat distribution in the granite was calculated from the electric field and the value of resistors. The temperature change was calculated from the heat distribution and the values of thermal resistance in the granite. The results of each calculated parameter for 20 kV (100% rise) and 10 kV (50% rise) are summarized in Figure 6. Table 6 shows the maximum and minimum values for each result. All of the maximum values in Table 6 were found in the small block in contact with the upper/HV electrode, and the values decreased as they moved away from the upper/HV electrode, with the minimum value found in the area in contact with the ground electrode.

Figure 6c shows the electric field distribution at the moment when 50% rise voltage was applied, and Figure 6d shows the electric field distribution at the moment when 100% rise voltage was applied. As mentioned above, the electric field distribution was high near the upper/HV electrode. In addition, the electric field distribution was found lower inside the plagioclase compared to other minerals. In Figure 6d, for example, plagioclase placed in the area defined by the y-z coordinates (4,5), (4,6) (5,6), (5,5) showed a lower electric field of plagioclase compared to their surroundings. Plagioclase has a higher void volume than other minerals (1.8 > 0.9%) and contains more water in the ED method (see Table 3) and that can be the reason behind the lower electric field distribution in plagioclase. In this study, the initial temperature of the minerals was set to 20 °C, so the electrical conductivity of the mineral-water composite was highly dependent on the water content. In other words, as the amount of water increases, the electrical conductivity increases (Equation (8)). Since electrical conductivity and resistivity have an inverse relationship, the resistance decreases, and thus, the electric field decreases according to Equation (3). This might be the reason for the low electric field inside plagioclase. When the resistance of the minerals adjacent to the plagioclase is lower than that of plagioclase, the voltage of the plagioclase with lower resistance is inevitably lower than that of the surrounding area because the current flowing in the circuit branches in the vertical direction is the same. Since the electric field is calculated by dividing the potential difference between two adjacent contacts by the distance between the two contacts, the electric field of the plagioclase naturally becomes lower than the surrounding area, while a higher electric field appears in the surrounding area. In comparing 50% between 100% rise of lightning impulse voltages (Figure 6c 50% rise vs. Figure 6d 100% rise), the latter case shows a clearer difference in the electric field distribution possibly due to the change of the electric field distribution in the granite depending on the magnitude of the applied voltage. It can be due to the difference in the dielectric breakdown electric field between water and rock (minerals) and its threshold duration of the HV wavefront is about 500 ns [9]. In other words, below 500 ns, the current flows inside the minerals, otherwise it flows through the water and not through the minerals.

Figure 6e,f shows the heat distribution at the moment of applying 50% rise voltage and 100% rise voltage, respectively. We can see the change in the heat generated in the granite from Figure 6e,f calculated from Equation (13). The heat generated in the upper part of the granite was higher than that of the lower part and was similar to the electric field distribution. There was almost no heat generated in the bottom parts in granite from our simulations, especially for the 50% rise of lightning impulse voltage (Figure 6e). As shown in Equation (13), the amount of heat generated is determined by the electric field distribution and the value of resistance, and since it is known from Figure 6c,d that the resistance of plagioclase is small due to its high electric conductivity, the electric field is low. Figure 6g,h shows the temperature change calculated at 50% rise and 100% rise of lightning impulse voltage. The maximum temperature change was about 3182 K near the upper/HV electrode, but the temperature change near to the ground electrode was almost zero. In particular, the plagioclase present in the area defined by the y-z coordinates (4,5), (4,7) (6,7), (6,5) showed less heat generation than their surrounding areas which showed a relatively high heat generation compared to plagioclase. The results showed that the temperature change differs with minerals and distance from the upper/HV electrode. By comparing Figure 6e (50% rise) with Figure 6f (100% rise), the temperature distribution is more heterogeneous in the latter case and affected by the different mineral placement that can deviate the electric field distribution (Figure 6d). Again, this temperature distribution difference can be the difference in the dielectric breakdown electric field as in the case of the electric field distribution.

As seen in Figure 6g,h, the temperature change is larger in the vicinity of the upper/HV electrode, indicating that the temperature change was occurring more likely by the heat generated due to the electric field and the resistance value of each mineral rather than by the heat resistance. The results obtained in this study reproduced similar trends as reported in other research works where the calculation was performed by considering only the electrical conductivity of minerals [12,13].

Walsh et al. (2013) used Voronoi tessellation to simulate granite in order to understand the thermal spallation. Their works using the Euler–Godunov model showed that the heat distribution differs in the boundary lines of the Voronoi tessellation (i.e., minerals). Our simulation works also showed that in spite of electric field distribution differences in the granite from their work, the heat variation was very small (see Figure 6f) [19].

Walsh et al. (2020) described simulation results with a 10-cm wide granite model with thickness of 2 cm. We used the same mineral composition in the granite as they did. They performed electric pulse crushing simulation using the ED method with needle electrodes placed at the center of the granite model for both the upper/HV and ground electrodes in 2D [12]. In addition, a high electric field was observed around the electrodes. In comparison with our results, they also observed a decrease in the electric field at the plagioclase and high electric field distribution near the electrodes [12]. In our work, we used both electric and dielectric parameters to understand the behavior of granite under HV electric pulse application. From our work, as we described before, the electric properties affected the electric distribution and heat generated in the minerals. At the same time, by considering dielectric constant in our work, the reason of the generated heat difference in different minerals was clear. On the other hand, Walsh et al. (2020) used 10 times larger input voltage compared to our work in order to simulate the thermal differences. If we consider the economic side of the mineral liberation by electric pulses, our method can be used to understand electrical and thermal properties of minerals under a lower input voltage application. Within this framework, further simulation works are to be carried out to confirm the heat difference under larger HV applications.

An application example of the ED method was described in the literature where it was applied on several minerals (i.e., quartz, pyrite, calcite, albite) which were embedded in cement and their selective liberation behavior was observed [19]. The size of the cemented minerals was 10 mm × 10 mm × 8 mm and the HV electrode was a needle electrode and flat electrode was placed as a grounding electrode. Channels were formed near the contact point of the electrodes, and crushing occurred when the electric pulse was applied on the cemented minerals [21]. The formation of channels and their connections which led to crushing occurred near the needle of the electrode and was due to the high electric field generated by the electric pulse. For the sample of quartz and cement, crushing occurred along the boundary between quartz and cement. The electric field distribution changed at the boundary of cement and quartz due to the difference in their electric conductivities, and this phenomenon is similar to the relationship between plagioclase and other minerals found in our study (Figure 6d,f). The electric field data obtained in this study are consistent with the general electric-pulse comminution theory, which states a difference in electric field occurs at boundaries with large differences in electric conductivity and dielectric constant, causing rock crushing due to rapid expansion of the discharge channels and associated heat generation [2]. The above comparison with the literature indicated that our simulation results showed the same/similar trend to the previous simulation and experiments of applying ED method on rocks, and thus, our methods and results obtained in this study are considered valid.

## 4. Conclusions

In this study, we investigated the electrical and thermal characteristics of granite under the application of electric pulses. An equivalent circuit model of a granite in 3D (1 mm × 1 mm × 1 mm) was developed. The electric field in the circuit was then calculated considering an electric pulse application. The heat and temperature changes generated in the minerals with the granite were calculated from that result. Until now, existing studies used either electrical conductivity or dielectric constant, but there has been no model that considers both. In this study, however, we used both parameters in the equivalent circuit model, calculated the values of the electric field distribution, and then, the heat distribution, and created the temperature change distribution from the heat distribution. This has not been performed before, and can lead to more detailed data under relatively low applied voltage by considering both parameters; so, the findings of this study can be useful for energy efficient rock crushing. The results of this study are summarized as follows:(1)The electric field of the granite confirmed that the electric field dropped near the plagioclase;(2)The electrical conductivity of the plagioclase was higher than that of the other minerals due to its higher void volume, which has a significant effect on the electric field distribution;(3)The temperature change in the granite observed a very high temperature change near the upper/HV electrode, but it was very small as away from that electrode.

## Figures and Tables

**Figure 1 materials-15-01039-f001:**
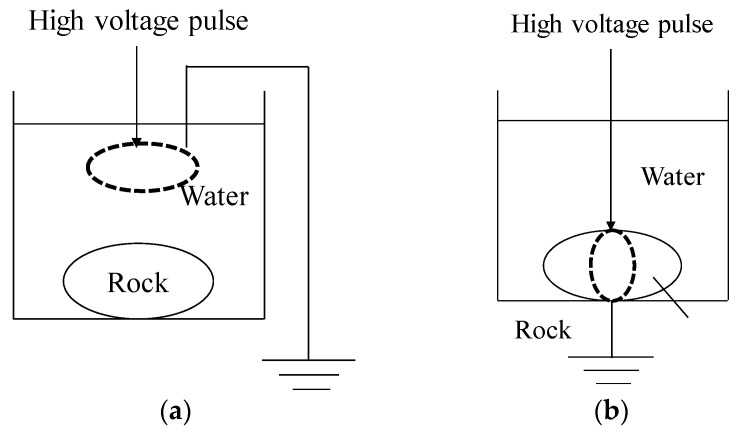
Model diagrams of different electric pulse crushing [4]. (**a**) EHD method; (**b**) ED method.

**Figure 2 materials-15-01039-f002:**
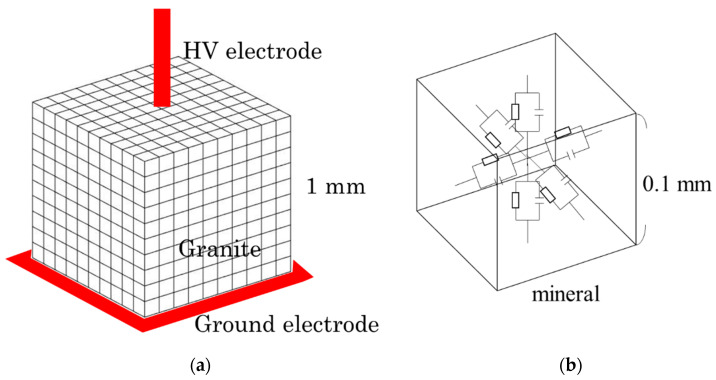
Concept of equivalent circuit model. (**a**) Granite; (**b**) equivalent circuit model for a small divided cube of granite.

**Figure 3 materials-15-01039-f003:**
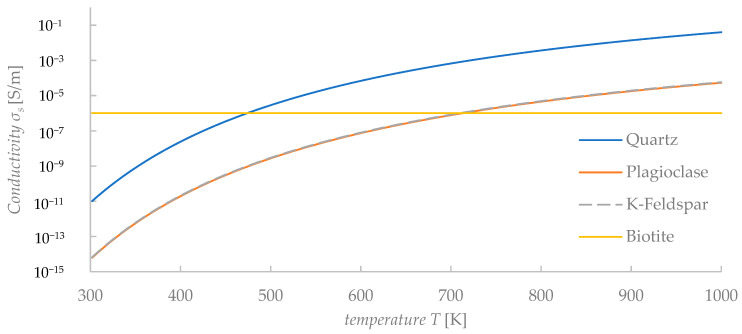
Calculated values of conductivity of minerals versus temperature.

**Figure 4 materials-15-01039-f004:**
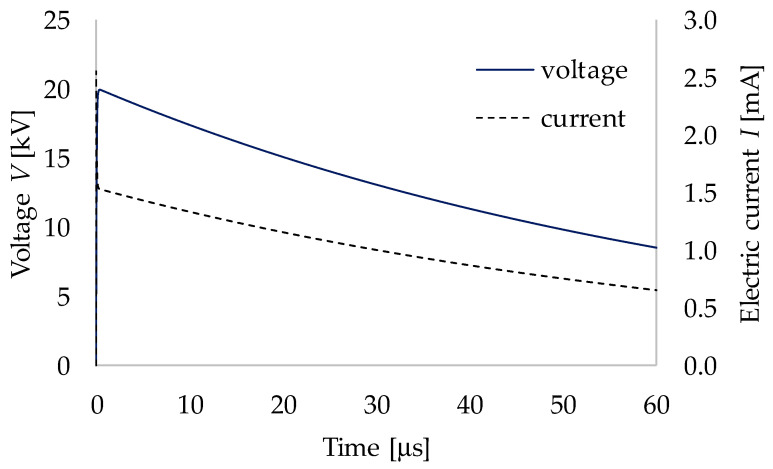
Applied voltage used in this simulation work and electric current.

**Figure 5 materials-15-01039-f005:**
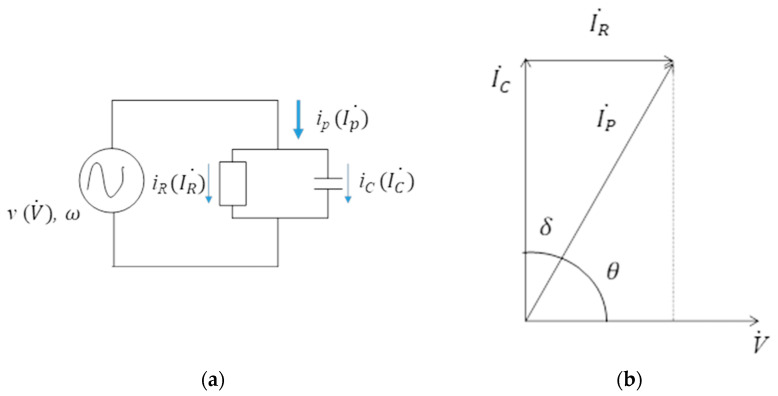
Equivalent circuit model for typical dielectric material. (**a**) Equivalent circuit; (**b**) vector analysis of the circuit.

**Figure 6 materials-15-01039-f006:**
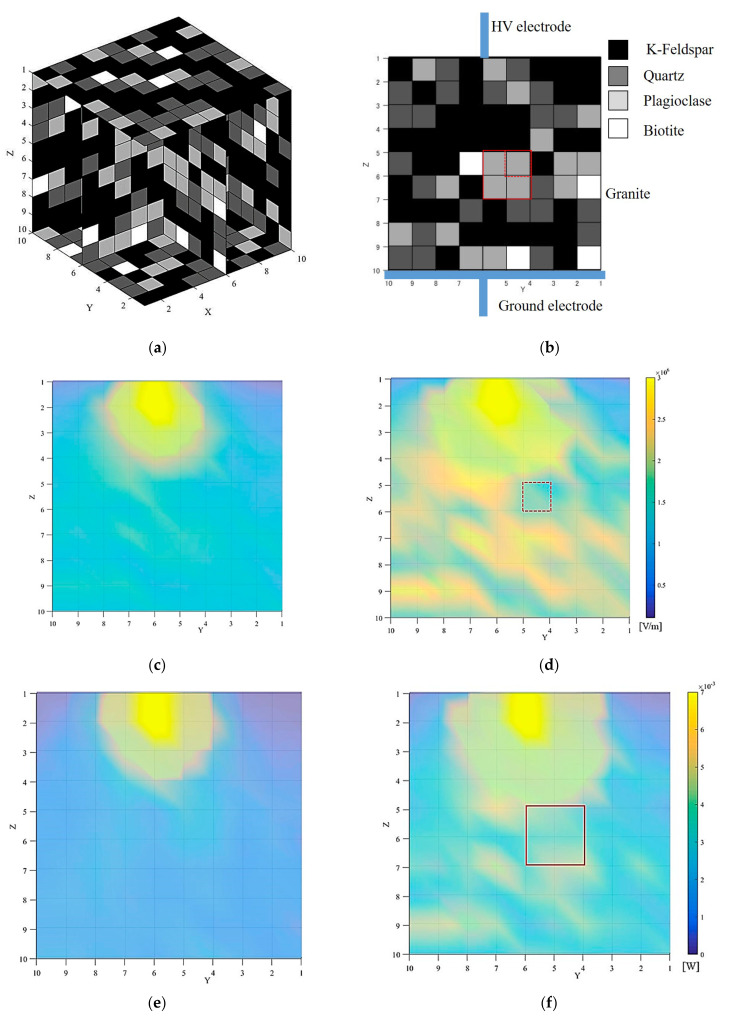
Simulation result when time changes. The red boxes in (**b**,**d**,**f**) are added for the visual aid. (**a**) Mineral distribution; (**b**) cross-section surface viewed from y-z direction; (**c**) electric field (50% rise of lightning impulse voltage); (**d**) electric field (100% rise of lightning impulse voltage); (**e**) heat (50% rise of lightning impulse voltage); (**f**) heat (100% rise of lightning impulse voltage); (**g**) variation of temperature (50% rise of lightning impulse voltage); (**h**) variation of temperature (100% rise of lightning impulse voltage).

**Table 1 materials-15-01039-t001:** Mineral composition of a typical granite [12].

Mineral	Volume [%]
Quartz	30
Plagioclase	20
K-Feldspar	45
Biotite	5

**Table 2 materials-15-01039-t002:** Values of the empirical parameters [12].

Mineral	Log(*A*) [log(s/m)]	*B* [eV]
Quartz	6.3	0.82
Plagioclase	0.041	0.85
K-Feldspar	0.11	0.85
Biotite	−13.8	0.00

**Table 3 materials-15-01039-t003:** Values of void volume * and permittivity of minerals in granite [12,13].

Mineral	Void [%]	Permittivity
Quartz	0.9	6.53
Plagioclase	1.8	6.91
K-Feldspar	0.9	6.2
Biotite	0.9	9.28

* Volume filled with water during ED method application.

**Table 4 materials-15-01039-t004:** Calculated values of resistance *R*, capacitance *C* and tan*δ* of minerals in granite.

Mineral	Resistance *R* [Ω]	Capacitance *C* [F]	tan*δ*
Quartz	1.57 × 10^7^	2.90 × 10^−15^	175
Plagioclase	5.54 × 10^6^	3.08 × 10^−15^	467
K-Feldspar	1.57 × 10^7^	6.15 × 10^−15^	82.5
Biotite	1.57 × 10^7^	8.26 × 10^−15^	61.4

**Table 5 materials-15-01039-t005:** Heat characteristics of minerals in granite [18].

Mineral	Heat Resistance [K/W]	Heat Conductivity [W/m⋅K]
Quartz	1300	7.69
Plagioclase	4673	2.14
K-Feldspar	4329	2.31
Biotite	4950	2.02

**Table 6 materials-15-01039-t006:** Maximum and minimum values for each result.

Voltage	Maximum Value of Electric Field [V/m]	Minimum Value of Electric Field [V/m]	Maximum Value of Heat [W]	Minimum Value of Heat [W]	Maximum Value of Temperature Change [K]	Minimum Value of Temperature Change [K]
50% rise	$17.7{\;\times \;10}^{6}$	$106{\;\times \;10}^{3}$	0.262	0.00	1133	0.0283
100% rise	$29.7{\;\times \;10}^{6}$	$104{\;\times \;10}^{3}$	0.7351	0.00	3182	0.0664

## Data Availability

Not applicable.

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
