# Peer review of "Simulation of Electrical and Thermal Properties of Granite under the Application of Electrical Pulses Using Equivalent Circuit Models"

_materials, 2022, doi:10.3390/ma15031039_

Round 1

Reviewer 1 Report

The authors report a simulation works for the electrical and thermal properties of Granite under a 20kV electrical pulse using equivalent circuit model. Some representations and discussions in the manuscript is not clear as:

1. What is the physical reason for choosing the cube size 0.1mm in the model? Please discuss the size/shape of minerals in granite.

2. It is not usual to calculate the electric field strength in the vertical direction only in the model. (L108-109) Could this give correct results?

3. Please show the features about this work and give references in others similar modeling works in introduction.

4. As the joule heat was generated from voltage and current, please show the corresponded current with same time scale in Fig.4. What is the time for the current to reach its maximum value?

5. With the dielectric material model (L181), please show and discuss the quantity of R, C and dielectric loss tangent for mineral unit(i.e.Quartz, Plagioclase, K-Feldspar, and Biotite ) respectively.

6. Is the basic simulation unit in this work 0.1mm x 0.1mm x 0.1mm as shown in FIg.2(b)? The spacial resolution for the electric field and current in the simulation shown in Fig.6 (c)-(h) is less than 0.1mm. Please show the quantity and origin of the spacial resolution in Fig.6 (c)-(h).

7.The unit of Fig.6 (c)-(h) should be labeled clearly on the figure.

8. Could a 1mm thick granite sustain 20kV applied? Please shows the meaning to calculate the heat flow and temperature distribution for a 1mm thick granite with 20kV applied. 

Author Response

Dear Madam/Sir,
It is our great honor for you to review our paper (materials-1559305). We would like to express our gratitude to you for your kind suggestions and comments. We have corrected our paper according to your kind suggestions. Please see the attached file. We will be highly grateful if you are kindly review it again.

Yours Sincerely,
Mahmudul Kabir
Akita University

Reviewer 2 Report

  1. A simple experimental study to validate the electric field and temperature measurement on the granite sample will be more appropriate for the manuscript.
  2. In figure -6 units related to the color graph for temperature, electric field, and heat are not mentioned  
  3. it will interesting to discuss the effect of high voltage generation from pyroelectric crystal during environmental temperature changes and effect of using high temperature lasers in relation to the reported work with following citations: a) https://www.sciencedirect.com/science/article/pii/S0272884219334558  b) https://ieeexplore.ieee.org/abstract/document/7808451 

Author Response

(The authors gave the same response as above.)

Reviewer 3 Report

This is a very good piece of work, well motivated (enhancing the efficiency in comminution), and well executed as theoretical, modeling and simulation work. It is very original and well presented, clearly written and with the results well supporting the drawn conclusions. It would be nice in the future to have more experimental data to validate the model. But at the moment the Authors have collected all experimental results in the literature I have been able to find. The paper is fine as it is.

Author Response

Dear Madam/Sir,

It is our great honor for you to review our paper (materials-1559305). We would like to express our gratitude to you for your kind comments. Your kind comments inspire us to proceed in further works on this subject.

Yours Sincerely,

Mahmudul Kabir

Akita University

Round 2

Reviewer 1 Report

The manuscript is improved and acceptable in present form.